# Anti-Psoriatic Effect of *Rheum palmatum* L. and Its Underlying Molecular Mechanisms

**DOI:** 10.3390/ijms232416000

**Published:** 2022-12-15

**Authors:** Ly Thi Huong Nguyen, Sang-Hyun Ahn, Heung-Mook Shin, In-Jun Yang

**Affiliations:** 1Department of Physiology, College of Korean Medicine, Dongguk University, Gyeongju 38066, Republic of Korea; 2Department of Anatomy, College of Korean Medicine, Semyung University, Jecheon-si 27136, Republic of Korea

**Keywords:** psoriasis, imiquimod, *Rheum palmatum* L., rhein, emodin, network pharmacology

## Abstract

Psoriasis is a chronic, immune-mediated inflammatory skin disorder. *Rheum palmatum* L. is a common traditional medicinal herb with anti-inflammatory and immunomodulatory activities. This study aimed to investigate the anti-psoriatic effects of the ethanolic extract from *R. palmatum* L. (RPE) and its chemical constituents, as well as the mechanisms underlying their therapeutic significance. An imiquimod (IMQ)-induced psoriasis-like mouse model was used to examine the anti-psoriatic effect of RPE in vivo. Network pharmacological analysis was performed to investigate the potential targets and related pathways of the RPE components, including rhein, emodin, chrysophanol, aloe-emodin, and physcion. The anti-inflammatory effects and underlying mechanisms of these components were examined using in vitro models. Topical application of RPE alleviated psoriasis-like symptoms and reduced levels of inflammatory cytokines and proliferation markers in the skin. Network pharmacological analysis revealed that RPE components target 20 genes that are linked to psoriasis-related pathways, such as IL-17, MAPK, and TNF signaling pathways. Among the five components of RPE, rhein and emodin showed inhibitory effects on TNF-α and IL-17 production in EL-4 cells, attenuated the production of CXCL8, CXCL10, CCL20, and MMP9, and reduced proliferation in HaCaT cells. Chrysophanol, aloe-emodin, and physcion were less effective than rhein and emodin in suppressing inflammatory responses and keratinocyte proliferation. The effects of these compounds might occur through the inhibition of the ERK, STAT3, and NF-κB signaling pathways. This study suggested the anti-psoriatic effect of RPE, with rhein and emodin as the main contributors that regulate multiple signaling pathways.

## 1. Introduction

Psoriasis is a chronic inflammatory skin disorder characterized by erythematous, thickened, and scaly plaques that can appear on the elbows, knees, scalp or any area of the body surface [1]. Psoriasis is a multifactorial disorder caused by complex interactions between genetic, immunological, and environmental factors [2]. Several studies have suggested the role of T helper (Th) cells, including Th1 and Th17 cells, in the development of this disease [3]. Th1 cells differentiate from naïve T cells in the presence of interleukin (IL)-12 to express interferon (IFN)-γ and tumor necrosis factor (TNF)-α, while IL-23 induces Th17 cell differentiation, resulting in the production of various cytokines including TNF-α, IL-6, IL-17, IL-21, and IL-22 [3,4]. The secretion of these inflammatory factors promotes keratinocyte hyperproliferation to induce epidermal hyperplasia and contributes to the chronic status of skin inflammation in psoriasis [5].

Topical treatment with corticosteroids is the most common therapeutic strategy for psoriasis; however, the long-term application of these medications may result in numerous adverse effects, including skin atrophy, impaired wound healing, and immune dysfunction [6]. Herbal medicines have attracted attention as new candidates for treating psoriasis due to their potential efficacy and safety [7]. *Rheum palmatum* L., also known as rhubarb, has been traditionally used to treat numerous diseases for thousands of years due to its anti-inflammatory, antimicrobial, and immunomodulatory activities [8]. Several clinical studies have demonstrated that the topical application of herbal formulas containing *R. palmatum* L. exhibited therapeutic effects against psoriasis by alleviating the disease severity [9,10]. An in vitro study also suggested that SIRB-001, a polyherbal formula consisting of *R. palmatum* L., *Lonicera japonica*, and *Rehmannia glutinosa* Libosch, exerts anti-psoriatic effects by promoting apoptosis, inhibiting proliferation and inflammation in keratinocytes, and by suppressing the production of IL-17 and IL-23 in immune cells [11]. An ethanolic extraction of *R. palmatum* L. significantly inhibited lipopolysaccharide (LPS)-induced IL-1β production in RAW264.7 macrophages [12]. In addition, rhubarb or *R. palmatum* L. has been considered a blood stasis-breaking medicine in traditional Chinese medicine (TCM) [13,14]. According to TCM theory, blood stasis is one of the main syndromes in psoriasis, and removing blood stasis and promoting blood circulation are treatment principles of TCM for psoriasis [15,16,17]. Blood stasis is still a developing concept but this concept includes vascular obstruction, abnormal blood flow, blood congestion, and blood contamination [18]. These results suggest the potential therapeutic benefits of this herb in psoriasis treatment. However, the mechanism underlying the anti-psoriatic effects of *R. palmatum* L. has not yet been investigated.

In this study, imiquimod (IMQ)-treated mice were used as a model to examine the anti-psoriasis effects of ethanolic extract from *R. palmatum* L. (RPE) in vivo. Network pharmacology was used to investigate the potential underlying mechanisms of RPE components (rhein, emodin, chrysophanol, aloe-emodin, and physcion) in psoriasis treatment. The network pharmacology results were further validated via in vitro experiments using HaCaT keratinocytes and EL-4 T cells.

## 2. Results

### 2.1. Effects of RPE on Psoriasis-like Symptoms in IMQ-Induced Mice

We initially investigated the effects of RPE on psoriasis in an IMQ-induced psoriasis mouse model. IMQ cream (5%) was applied to the back skin of the mice for six consecutive days (Figure 1A). RPE (0.1% and 1%) or vehicle was topically treated 1 h before IMQ stimulation. IMQ mice exhibited the pathological features of psoriasis, whereas treatment with RPE significantly inhibited these symptoms (Figure 1B). RPE treatment also reduced the Coscam scores, including curvature, keratin, and pigmentation, when compared to the IMQ group (Figure 1C). The psoriasis area and severity index (PASI) scoring system was used to assess psoriasis severity, including thickness, erythema, and scaling. Mice treated with RPE exhibited lower PASI scores than those treated with IMQ (*p* < 0.001) (Figure 1D). The effects of RPE on psoriasis-like symptoms were comparable with the positive control, clobetasol (CP). IMQ application significantly decreased body weight when compared to the normal control (NC) group (*p* < 0.001); however, this decrease was reversed by treatment with RPE 1% (*p* < 0.05) but not CP (Figure 1E). RPE treatment did not affect the IMQ-induced increase in the spleen index, while CP significantly reduced the spleen index in IMQ-treated mice (*p* < 0.05) (Figure 1F).

### 2.2. Effects of RPE on Epidermal Hyperplasia and the Expression of Proliferation Markers and Inflammatory Cytokines in IMQ-Induced Mice

H&E staining revealed that the IMQ mice had significant epidermal hyperplasia. In contrast, topical application of 0.1% and 1% RPE significantly decreased epidermal thickness compared to that in the IMQ group (*p* < 0.01) (Figure 2A). In addition, the levels of the proliferation markers K16 and PCNA were also alleviated by treatment with RPE (*p* < 0.05) (Figure 2B). The positive control, CP treatment, also decreased epidermal hyperplasia and levels of K16 and PCNA in skin lesions (*p* < 0.001) (Figure 2A,B). Th1 and Th17 cells and their cytokines contribute to skin inflammation and epidermal hyperplasia in psoriasis [3]. We hypothesized that RPE exerts its anti-psoriatic effects by inhibiting Th1/Th17 cytokines. We found that the levels of IL-17, IL-6, TNF-α, and IFN-γ were significantly elevated in the IMQ group, whereas the production of these cytokines was suppressed by RPE application (*p* < 0.05) (Figure 2C). In addition, IL-22 levels were reduced by RPE. The involvement of VEGF has also been reported in keratinocyte hyperplasia in psoriasis [19]. In this study, IMQ upregulated the skin level of VEGF, which was significantly decreased by RPE (*p* < 0.01) (Figure 2C). The effects of RPE on the expression of inflammatory cytokines in IMQ-induced mice were similar to the positive control, CP (Figure 2C).

### 2.3. Quantitative Assessment of the Chemical Composition of RPE

The constituents of RPE were detected using high-performance liquid chromatography (HPLC) analysis, and chromatograms of the standard compounds and RPE are shown in Figure 3B,C, respectively. The retention times of aloe-emodin, rhein, emodin, chrysophanol, and physcion were 7.467 min, 7.95 min, 10.911 min, 13.716, and 15.05 min, respectively. The concentrations of these five compounds in RPE were 0.969–6.426 mg/g and rhein was the compound with the highest concentration in the extract (Figure 3A).

### 2.4. Compound–Target and Protein–Protein Interaction (PPI) Network Construction

Network pharmacology was performed to investigate potential anti-psoriatic targets and mechanisms of RPE compounds. We identified 50 targets of the active compounds in RPE using the traditional Chinese medicine systems pharmacology database and analysis platform (TCMSP) database (Appendix A). A total of 1308 psoriasis-related targets were retrieved from the DisGeNET database (disease ID: C0033860). The targets of the RPE components and psoriasis-related targets were merged to obtain common targets. A Venn diagram of this intersection is shown in Figure 4A. Twenty common targets of RPE compounds and psoriasis were identified and these targets were tagged as potential anti-psoriatic targets for RPE components. Cytoscape 3.8.2 was used to visualize the compound-common target network. As shown in Figure 4B, the network contained 25 nodes and 37 edges. The nodes represent common targets and compounds, while the edges represent the compound–target interaction.

A PPI network of 20 potential anti-psoriatic targets of RPE components was constructed using the Search Tool for the Retrieval of Interacting Genes/Proteins (STRING) database; this contained 20 nodes and 44 edges (Figure 4C). The analysis of node degrees is shown in Figure 4D. Several targets, such as TP53 (17), EGF (16), and TNF (16), exhibited higher values, implying that these targets play various important roles in the therapeutic effects of RPE components.

### 2.5. Gene Ontology (GO) and Kyoto Encyclopedia of Genes and Genomes (KEGG) Enrichment Analysis

GO and KEGG enrichment analyses for potential targets were performed using the Enrichr tool. Figure 5A and Appendix A show the top ten terms of the three categories (Biological Process, Cellular Component, and Molecular Function) of the GO functional enrichment analysis. According to these results, potential targets were significantly involved in the response to cytokines, regulation of transcription, and modulation of protein phosphorylation, which likely play a role in the progress of psoriasis. We also performed the KEGG enrichment analysis to investigate the potential signaling pathways of the targets (Figure 5B and Appendix A). The targets of the RPE components were mainly enriched in the IL-17 signaling pathway, TNF signaling pathway, MAPK signaling pathway, cell cycle, and Th17 differentiation, which are all important pathways in the pathogenesis of psoriasis. The potential targets involved in the IL-17 signaling pathway are indicated in blue in Figure 5C.

### 2.6. Effects of RPE Compounds on the Inflammatory Response in PI-Stimulated EL-4 Cells

Network pharmacology analysis showed that RPE compounds targeted TNF and IL-17 signaling pathways; here, in vitro experiments were conducted to examine whether these compounds have any effect on the production of TNF-α and IL-17 in activated T cells. The XTT results indicated that RPE compounds (rhein, emodin, chrysophanol, aloe-emodin, and physcion) at a concentration of 10 μM did not show any cytotoxic effects on the viability of EL-4 cells (a murine T cell line) (Figure 6A); hence, this concentration was used for further experiments. EL-4 cells stimulated with PI (10 ng/mL PMA and 100 ng/mL ionomycin) exhibited a significant increase in TNF-α and IL-17 production (*p* < 0.05). In contrast, as shown in Figure 6B, pretreatment with rhein or emodin significantly suppressed the levels of TNF-α and IL-17 in PI-stimulated EL-4 cells (*p* < 0.05). Aloe-emodin only suppressed TNF-α secretion (*p* < 0.05). The inhibitory effects of these compounds on cytokine production were comparable with the positive control, clobetasol. Chrysophanol and physcion did not inhibit either TNF-α or IL-17 production in EL-4 cells (Figure 6B).

### 2.7. Effects of RPE Compounds on the Inflammatory Response and Proliferation in M5-Stimulated HaCaT Cells

M5 (a cocktail of TNF-α, IL-1α, IL-17, IL-22, and Oncostatin M, 10 ng/mL each) has been reported to mimic the microenvironment of psoriatic lesions [20]. Here, we investigated the effects of RPE compounds on chemokine production in M5-stimulated HaCaT keratinocytes. Figure 7A shows that a concentration of 10 μM for the RPE compounds (rhein, emodin, chrysophanol, aloe-emodin, and physcion) did not cause cytotoxicity in HaCaT cells; hence, this concentration was used for further experiments. As shown in Figure 7B, pretreatment with rhein significantly suppressed the M5-induced upregulation of CXCL8, CXCL10, CCL20, and MMP9 in HaCaT cells (*p* < 0.05). Emodin reduced the production of CXCL8, CXCL10, and MMP9 (*p* < 0.01). Chrysophanol decreased the production of CXCL10 and MMP9 (*p* < 0.05), whereas physcion only reduced CXCL10 secretion (*p* < 0.01), and aloe-emodin only suppressed MMP9 production (*p* < 0.01) in M5-treated cells. These compounds showed comparable effects with clobetasol (the positive control) on CXCL8 and CXCL10 production but superior inhibitory effects on CCL20 and MMP9 production. Moreover, rhein and emodin significantly reduced the percentage of Ki67(+) cells in M5-treated HaCaT cells (*p* < 0.05), suggesting the antiproliferative effect of these compounds (Figure 7C).

### 2.8. Effects of RPE Compounds on the Activation of the ERK, STAT3, and NF-κB Signaling Pathways in M5-Stimulated HaCaT Cells

To investigate the underlying mechanism that mediates the anti-psoriatic effect of the RPE components, the effects of these compounds on the ERK (extracellular signal-regulated kinase), STAT3 (signal transducer and activator of transcription 3), and NF-κB (nuclear factor kappa B) signaling pathways in M5-stimulated HaCaT cells were examined. As shown in Figure 8A, pretreatment with rhein significantly suppressed the M5-induced phosphorylation of ERK and STAT3 (*p* < 0.05). Emodin only reduced STAT3 phosphorylation in HaCaT cells (*p* < 0.05), whereas chrysophanol, aloe-emodin, and physcion did not downregulate the phosphorylation of ERK and STAT3 in HaCaT cells. All five compounds suppressed the NF-κB activation by decreasing the NF-κB level in the nucleus and increasing the IκB-α level in the cytoplasm (*p* < 0.05) (Figure 8B). The effects of these compounds on ERK, STAT3, and NF-κB signaling pathways were similar to the positive control, clobetasol.

## 3. Discussion

Psoriasis is a common immune-mediated inflammatory skin disorder that has detrimental effects on the quality of life of patients [21]. Topical application of corticosteroids, such as dexamethasone or clobetasol propionate, is the most common therapeutic strategy for the management of psoriasis; however, these drugs can cause various systemic and cutaneous negative effects, such as immunosuppression, dermal atrophy, and delayed wound healing, which leads to low satisfaction among patients with psoriasis [6]. Traditional herbal medicines have attracted attention as novel psoriasis therapy options [7,22]. This study demonstrated the anti-psoriatic activities of RPE, an ethanolic extract of the medicinal plant *Rheum palmatum* L. and investigated the underlying mechanisms that mediate its therapeutic effects.

IMQ-induced skin inflammation has been established in an animal model of psoriasis. IMQ-treated mice exhibit phenotypes similar to human psoriasis, including erythema, thickening, and scaling [23]. In this study, after six days of IMQ application, BALB/c mice also exhibited severe Coscam and PASI scores with a significant increase in epidermal thickness. In contrast, we demonstrated that psoriasis-like symptoms were alleviated by RPE treatment. This finding supports the results of a previous study showing that a 50% ethanolic extract of *R. palmatum* inhibited arachidonic acid-induced skin inflammation in mice [24]. Moreover, the decrease in body weight caused by IMQ application was recovered by 1% RPE treatment but not by the positive control drug (clobetasol propionate, CP). These results suggest that RPE possesses anti-psoriatic effects without any toxicity in the mice model. Unexpectedly, RPE treatment did not inhibit the IMQ-induced increase in the spleen index. This might be due to the fact that RPE was topically applied to the dorsal skin and might have only exerted local effects on psoriatic skin lesions but not significant effects on the spleen. However, CP treatment decreased the spleen index when compared with the IMQ group, and this was also remarkably lower than that in normal mice, indicating systemic immunosuppression, which is an adverse effect of corticosteroid treatment.

The infiltration of immune cells into skin lesions and cutaneous inflammation are typical characteristics of psoriasis. Naïve T cells can differentiate into Th1 and Th17 cells, migrate to the dermis, and produce typical proinflammatory cytokines. Th1 cells produce TNF-α and IFN-γ, whereas Th17 cells can produce IL-17, IL-6, TNF-α, IL-21, and IL-22. These cytokines can trigger and maintain chronic inflammation, as well as induce keratinocyte hyperproliferation, in psoriatic lesions [25,26]. IL-17 signaling has been demonstrated to be involved in the development of psoriasis in IMQ-induced mouse models [27]. In this study, IMQ application upregulated the levels of both Th1 and Th17 cytokines, including TNF-α, IFN-γ, IL-17, and IL-6, in the skin lesions. However, RPE treatment remarkably suppressed the production of these inflammatory cytokines. Network pharmacology analysis revealed that RPE compounds target genes related to IL-17, TNF, and Th17 differentiation pathways. This was verified in the in vitro experiments, which showed that rhein and emodin, two components of RPE, significantly inhibited the production of TNF-α and IL-17 in PI-stimulated EL-4 cells (a murine T cell line). VEGF is thought to play an important role in psoriasis pathogenesis through both autocrine and paracrine mechanisms. VEGF secreted by keratinocytes can directly stimulate the proliferation of these cells and induce angiogenesis to support epidermal hyperplasia [19]. In the current study, IMQ significantly increased the level of VEGF in the skin lesions, which was suppressed by RPE treatment. These results suggest that the anti-psoriatic effect of RPE is mediated by the inhibition of the production of inflammatory molecules in skin lesions.

Although psoriasis is believed to be an immune-mediated disease, keratinocytes also play an important role in its progression. Keratinocyte hyperproliferation is an important hallmark of psoriasis, leading to epidermal hyperplasia and psoriatic plaque formation [28]. In this study, epidermal thickness and the expression of keratinocyte proliferation markers PCNA and K16 were significantly increased by IMQ stimulation and reduced by treatment with RPE. Upon activation, keratinocytes proliferate excessively and produce various inflammatory chemokines to attract immune cells to psoriatic lesions [29]. For example, CXCL8 attracts neutrophils, CXCL10 recruits Th1 cells, or CCL20 induces the infiltration of Th17 cells into the skin [28,30]. Keratinocytes can also produce a variety of matrix metalloproteinases (MMPs), such as MMP2, MMP3, and MMP9, which are involved in the inflammatory responses, cell migration, and angiogenesis in psoriasis [31]. These inflammatory mediators are also the downstream molecules of the IL-17 signaling pathway, which is the main target pathway of RPE components. The effects of RPE compounds on the inflammatory response in keratinocytes were further validated using in vitro experiments. In this study, we used HaCaT keratinocytes stimulated with M5, a cocktail of five inflammatory cytokines (TNF-α, oncostatin M, IL-1α, IL-17A, and IL-22), as an in vitro model of psoriasis with typical upregulation in the production of cytokines, chemokines, and antimicrobial peptides [20,32]. Our results demonstrated that pretreatment with rhein and emodin suppressed the M5-induced increase in the secretion of CXCL8, CXCL10, CCL20, and MMP9 in M5-treated HaCaT cells. Chrysophanol, aloe-emodin, and physcion were less effective than rhein and emodin at reducing the production of these inflammatory molecules. Previous studies also reported that M5 stimulation could trigger keratinocyte proliferation in vitro by increasing the percentage of cells in the S phase and the expression of cyclin proteins [33,34]. Similarly, in the current study, Ki67 proliferation assays revealed that M5 treatment increased hyperproliferation in HaCaT cells, which was reduced by rhein and emodin pretreatment.

The underlying mechanisms that mediate the antipsoriatic effects of RPE were also explored. Accumulating evidence suggests that the ERK and STAT3 pathways play a crucial role in the pathogenesis of psoriasis by inducing keratinocyte proliferation and inflammation, and the levels of ERK and STAT3 in psoriatic lesions are also higher than those in normal skin [35,36]. Inhibition of STAT3 activation has shown efficacy in treating psoriasis, both in preclinical and clinical studies [36]. The topical application of JSI287, an ERK inhibitor, improved IMQ-induced psoriasis-like symptoms in mice [37]. NF-κB plays a crucial role in psoriasis by regulating inflammatory responses and cell proliferation [38]. Under normal conditions, inactive NF-κB binds to IκB-α in the cytoplasm. Upon stimulation, IκB-α is phosphorylated and degraded by IκB kinase (IKK), leading to the liberation and translocation of NF-κB into the nucleus to regulate the expression of inflammation-related targets [39]. NF-κB is highly expressed in psoriatic lesions and its inhibition might be a potential target for psoriasis treatment [40,41]. Network pharmacological analysis showed that RPE compounds target various genes involved in the MAPK, JAK/STAT, and NF-κB signaling pathways. In vitro results demonstrated that rhein and emodin can reduce M5-induced ERK and STAT3 activation in M5-treated HaCaT keratinocytes. Rhein, emodin, chrysophanol, aloe-emodin, and physcion suppressed NF-κB signaling by increasing the cytoplasmic level of IκB-α and reducing the nuclear level of NF-κB in M5-stimulated HaCaT cells. These findings suggest that the anti-psoriatic effect of RPE might occur through the suppression of the activation of the ERK, STAT3, and NF-κB pathways.

## 4. Materials and Methods

### 4.1. Plant Materials

*Rheum palmatum* L. was obtained from Kyung Hee University Hospital (Seoul, Republic of Korea) and verified by Professor In-Jun Yang (Dongguk University, Gyeongju, Republic of Korea). A voucher specimen (2017-A-09) was deposited at the College of Korean Medicine at Dongguk University. *Rheum palmatum* L. (15 g) was extracted with 70% aqueous ethanol (150 mL) at 70 °C for 3 h, after which the extract was filtered through Whatman no.2 filter paper (Whatman International, Maidstone, UK), concentrated using a rotary vacuum evaporator, and then freeze-dried (FD8508S, Busan, Republic of Korea) (yield 17.33% *w*/*w*).

### 4.2. Animal Grouping and Treatments

BALB/c (Bagg albino strain C) mice (6-week-old, male, and 18–20 g) were purchased from Koatech (Gyeonggi, Republic of Korea). All animal experiments were approved by the Institutional Animal Care and Use Committee of Dongguk University (Approval No. IACUC-2020-08). Imiquimod (IMQ) (Aldara 5% cream, 3M Health Care Limited, Loughborough, England) was used for psoriasis induction in mice. Clobetasol propionate (CP) (Sigma-Aldrich, St. Louis, MO, USA) was utilized as the positive control drug. The mice were assigned into five groups (*n* = 6 per group): normal control (NC), IMQ-treated (IMQ), 0.1% RPE, 1% RPE, and 0.05% CP. All mice were acclimatized for one week before the experiment. The backs of the BALB/c mice were shaved. To induce psoriasis-like skin lesions, IMQ cream (62.5 mg) was applied daily for six days on the back skin of the mice in the IMQ-, 0.1% RPE-, 1% RPE-, and CP-treated groups. All drugs were dissolved in acetone, as previously described [42], and topically applied for 1 h before IMQ application. Clinical severity was scored using the modified Psoriasis Area Severity Index (PASI) with assessments for thickness, scaling, and erythema. The measurement system included no symptoms (0), mild (1), moderate (2), severe (3), or very severe (4). Skin symptoms, including curvature, keratin, and pigmentation, were also examined using a Coscam USB225 Skin and Hair Analyzer (SOMETECH, Seoul, Republic of Korea), as previously described [43]. All the mice were sacrificed by isoflurane inhalation. The body and spleen weights of the mice were measured. The spleen index was defined as the ratio of the spleen weight to body weight. Dorsal skin lesions were collected for further analysis.

### 4.3. Histological Assessment

Back skin tissues were preserved in a 4% paraformaldehyde solution and embedded in paraffin blocks. The samples were cut into 5 μm thick sections and stained with hematoxylin and eosin (H/E). An automated microscope (Lionheart FX) and Gen5 software (Biotek, Winooski, VT, USA) were used to evaluate histological changes in skin lesions.

### 4.4. High-Performance Liquid Chromatography (HPLC) Analysis

Rhein (CFN99157, purity 98%), emodin (CFN98834, 99%), chrysophanol (CFN98751, 98%), aloe-emodin (CFN98749, 98%), and physcion (CFN98848, 98.5%) were obtained from ChemFaces (Hubei, China). RPE components were quantified using an HPLC 1290 system (Agilent, Santa Clara, CA, USA) at the Korea Basic Science Institute (Seoul, Republic of Korea). RPE (10 μL, 20 mg/mL) was separated on a Kinetex C18 column (4.6 × 250 mm, 5 µm, Phenomenex, Torrance, CA, USA) at a flow rate of 1 mL/min. Mobile phases A and B consisted of 0.1% phosphoric acid and acetonitrile, respectively. The solvent gradient components were as follows: 40–50% (B) for 50 min, 50–100% (B) for 20 min, and equilibration for 5 min. The column temperature was maintained at 25 °C. Rhein, emodin, chrysophanol, aloe-emodin, and physcion were detected at 278 nm wavelength. The concentrations of the compounds in RPE were calculated using standard curves.

### 4.5. Network Pharmacology

Network pharmacology was performed to identify potential targets and mechanisms of RPE components in psoriasis treatment. The Traditional Chinese Medicine Systems Pharmacology (TCMSP) database (https://old.tcmsp-e.com/tcmsp.php, accessed on 3 April 2021) [44] was used to retrieve the target genes of the RPE components. DisGeNET database (https://www.disgenet.org/, accessed on 3 April 2021) with the keyword “psoriasis” was used to access psoriasis-related genes [45]. The compound–target network was visualized using Cytoscape 3.8.2 (Cytoscape Consortium, San Diego, CA, USA) [46]. Protein–protein interactions (PPI) play a key role in the regulation of biological processes and the investigation of these PPIs might be helpful in determining potential targets for disease treatment [47]. In this study, a PPI network of overlapping targets of RPE compounds and psoriasis was constructed using the Search Tool for the Retrieval of Interacting Genes/Proteins (STRING) (https://string-db.org/, accessed on 3 April 2021) for multiple proteins in humans (*Homo sapiens*) [48]. The Enrichr tool (https://maayanlab.cloud/Enrichr/, accessed on 3 April 2021) was used for enrichment analysis with the Gene Ontology (GO) terms (Biological Process, Molecular Function, and Cellular Component) and the Kyoto Encyclopedia Genes and Genomes (KEGG) pathway database [49]. GO terms and KEGG pathways (*p* < 0.05) were visualized using the SRplot tool (http://www.bioinformatics.com.cn/, accessed on 3 April 2021).

### 4.6. Cell Culture

HaCaT cells (a human keratinocyte cell line) were kindly supported by the Korea Institute of Oriental Medicine (Daegu, Republic of Korea). EL-4 cells (murine T cell line) were purchased from the Korean Cell Line Bank (Seoul, Republic of Korea). HaCaT and EL-4 cells were grown in Dulbecco’s modified Eagle medium (Welgene, Gyeongsangbuk, Republic of Korea) supplemented with 10% fetal bovine serum and 1% penicillin/streptomycin (Invitrogen, Carlsbad, CA, USA) at 37 °C in a 5% CO_2_ humidified incubator. The cells were sub-cultured upon reaching 80–90% confluency.

### 4.7. Cell Viability

The effects of RPE components on the viability of HaCaT and EL-4 cells were determined using a Cell Proliferation Kit II (XTT) (#11465015001, Roche Diagnostics GmbH, Mannheim, Germany). After treating the cells with rhein, emodin, chrysophanol, aloe-emodin, or physcion (1, 10, and 50 µM) for 24 h, 50 µL of the XTT solution was added and incubated for 4 h. The optical density was evaluated at 450 nm (reference wavelength: 650 nm) using a microplate reader (Tecan, Männedorf, Switzerland).

### 4.8. Enzyme-Linked Immunosorbent Assay (ELISA)

Skin samples were lysed with a tissue extraction reagent (FNN0071, Bender MedSystems GmbH, Vienna, Austria) and centrifuged at 10,000× *g* for 20 min, followed by supernatant collection. The levels of TNF-α, IFN-γ, IL-6, IL-17, IL-22, and VEGF in the skin lysates were measured using ELISA kits, according to the manufacturer’s instructions. In the in vitro study, HaCaT cells and EL-4 cells were pretreated with rhein, emodin, chrysophanol, aloe-emodin, physcion, or clobetasol propionate (10 μM) for 1 h before treatment with M5 (TNF-α, oncostatin M, IL-1α, IL-17A, IL-22, 10 ng/mL each), PI (phorbol myristate acetate (PMA, 10 ng/mL) and ionomycin (100 ng/mL)) for 24 h. The levels of TNF-α and IL-17 in the culture supernatants from EL-4 cells and the levels of CCL20, CXCL8, CXCL10, and MMP9 in the culture supernatants from HaCaT cells were evaluated using ELISA kits, based on the manufacturer’s instructions. Absorbance was determined at 450–550 nm using a microplate reader (Tecan, Männedorf, Switzerland). Human CXCL8 (K0331216P), CXCL10 (K0331210P), and mouse TNF-α (K0331186P), IFN-γ (K0331138P), IL-6 (K0331230P), IL-17 (K0331268P), IL-22 (K0332144P), VEGF (K0331224P) ELISA kits were obtained from Koma Biotech Inc. (Seoul, Republic of Korea). Human CCL20 (DM3A00) and MMP9 (DMP900) ELISA kits were obtained from R&D Systems (Minneapolis, MN, USA).

### 4.9. Ki67 Proliferation Assay

The effects of RPE compounds on the proliferation of M5-treated HaCaT cells were evaluated using a Muse Ki67 proliferation kit (MCH100114, Luminex Corporation, Austin, TX, USA) based on the manufacturer’s instructions. The cells were grown in six-well plates (5 × 105 cells/well) for 24 h and then incubated with rhein, emodin, chrysophanol, aloe-emodin, physcion, or clobetasol propionate (10 μM) for 1 h before M5 treatment for another 24 h. HaCaT cells were then processed using fixation and permeabilization buffers, followed by incubation with Ki67-PE and IgG1-PE antibodies; the percentage of Ki67(+) cells was assessed using a Muse Cell Analyzer (Merck KGaA, Darmstadt, Germany).

### 4.10. Western Blot Analysis

Skin samples were lysed with a tissue extraction reagent and centrifuged at 10,000× *g* for 20 min, followed by supernatant collection. HaCaT cells were lysed using RIPA lysis buffer, which contained protease and phosphatase inhibitors (Atto, Tokyo, Japan). The cell lysates were centrifuged at 8000× *g* for 10 min and then the supernatants were collected. NE-PER extraction reagents (#78833, Thermo Fisher Scientific, Waltham, MA, USA) were used for the nuclear and cytoplasmic protein extractions from HaCaT cells, according to the manufacturer’s protocols. Protein concentration was measured using Bradford protein assay reagent (Bio-Rad, Hercules, CA, USA). Subsequently, the proteins (20–30 μg) were resolved by 10% SDS-PAGE and blotted onto PVDF membranes (IPVH00010, Millipore, Carrigtwohill, Ireland). The membranes were blocked in 5% skim milk for 2 h and then incubated with primary antibodies overnight followed by secondary antibodies. The primary antibodies used were p-ERK (#4370S), p-STAT3 (#9131S), ERK (#9102S), STAT3 (#9139S), NF-κB (#8242S), and IκB-α (#4814S) from Cell Signaling Technology (Danvers, MA, USA); lamin B2 (ab151735) and K16 (ab53117) from Abcam (Cambridge, UK); PCNA (sc-56) from Santa Cruz Biotechnology (Dallas, TX, USA); β-actin (A1978) from Sigma-Aldrich (St. Louis, MO, USA). The secondary antibodies included horseradish peroxidase (HRP)-conjugated goat anti-rabbit IgG antibody (#31460), Invitrogen (Carlsbad, CA, USA), and HRP-conjugated goat anti-mouse IgG antibody (ADI-SAB-100-J), Enzo Life Sciences (Farmingdale, NY, USA). The membranes were visualized using enhanced chemiluminescence reagents and a ChemiDoc system (Bio-Rad, Hercules, CA, USA). Protein band intensities were assessed using a GelPro analyzer (ver. 3.1, Media Cybernetics, Rockville, MD, USA).

### 4.11. Statistical Analysis

Statistical analyses were conducted using GraphPad Prism 9.0.0 (GraphPad Software, San Diego, CA, USA). The data are presented as mean ± standard deviation (SD) of at least three independent experiments. Differences between groups were analyzed using one-way ANOVA or Student’s *t*-test for unpaired experiments, and *p*-values < 0.05 were considered statistically significant.

## 5. Conclusions

In this study, biological research and network pharmacology were combined to demonstrate the anti-psoriatic effects of RPE. Topical application of RPE alleviated psoriasis-like symptoms in mice by suppressing the inflammatory response and epidermal hyperplasia. Network pharmacology analysis suggested that the five bioactive components of RPE target multiple genes and signaling pathways related to psoriasis. In vitro validation suggested that rhein and emodin may be the main contributors to the anti-psoriatic effects of RPE.

## Figures and Tables

**Figure 1 ijms-23-16000-f001:**
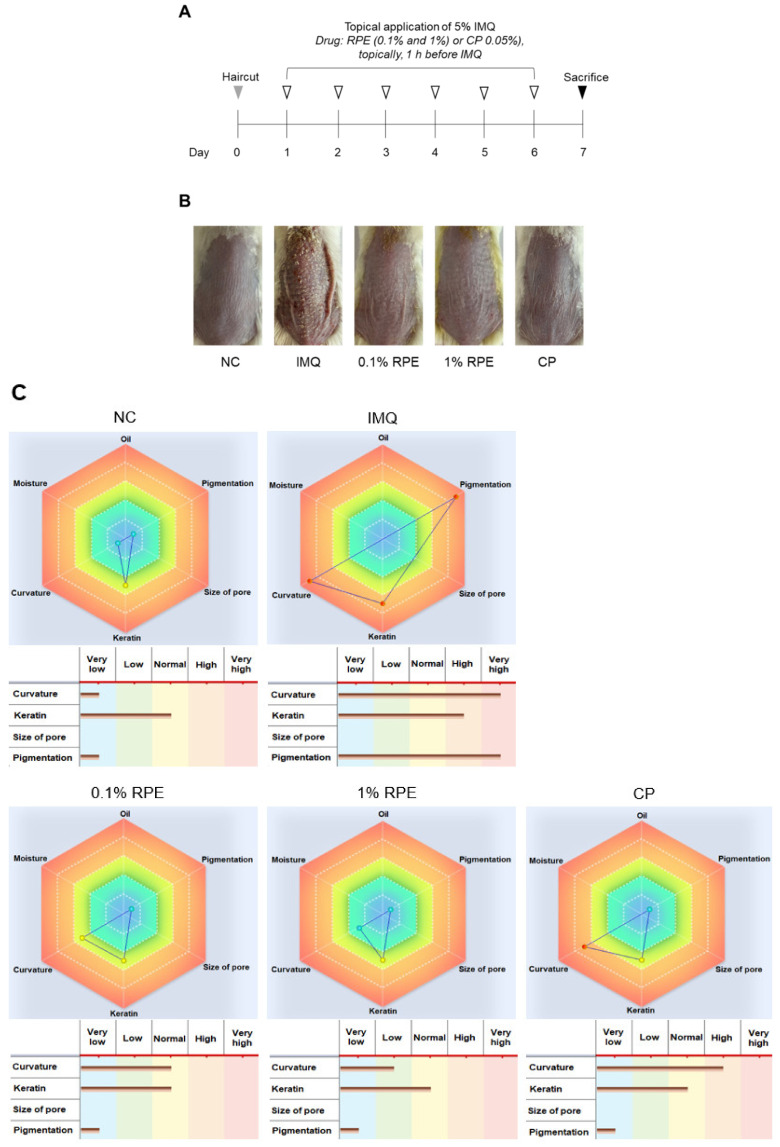
Effects of RPE on psoriasis-like symptoms in IMQ-induced mice. (**A**) Experimental schedule for IMQ-induced psoriasis model and treatments. (**B**) Representative images of skin lesions. (**C**) Skin analysis by CosCam. Blue: very low score, green: low score, yellow: normal score, light orange: high score, dark orange: very high score. (**D**) PASI score of skin lesions, including thickness, erythema, and scaling (0–4) and a total score (0–12). (**E**) Body weight of mice from five groups. (**F**) Spleen index was defined as the ratio of spleen weight and body weight. Data are expressed as means ± SDs (*n* = 6 per experiment). * *p* < 0.05, *** *p* < 0.001 vs. the IMQ group. NC: normal control, IMQ: imiquimod, RPE: *Rheum palmatum* L. extract, CP: clobetasol propionate, and PASI: Psoriasis Area Severity Index.

**Figure 2 ijms-23-16000-f002:**
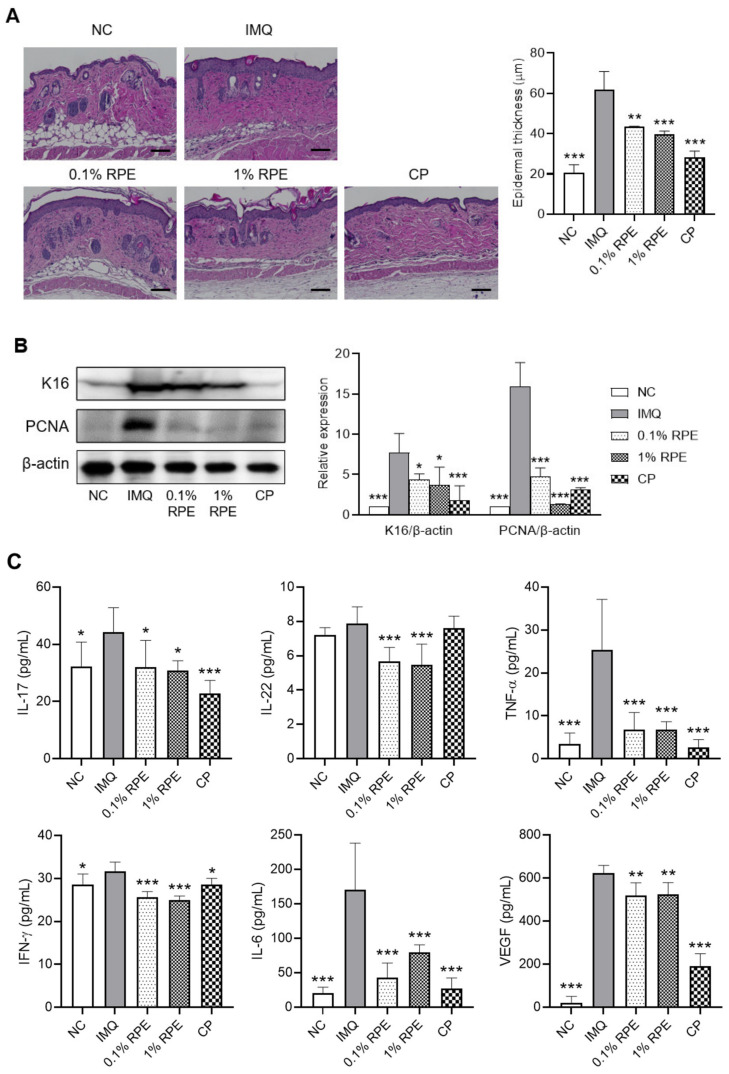
Effects of RPE on epidermal hyperplasia and the expression of proliferation markers and inflammatory cytokines in IMQ-induced mice. (**A**) H&E staining of skin lesions. Scale bar: 200 μm. Epidermal thickness was evaluated. (**B**) The levels of proliferation markers K16 and PCNA in skin lesions were measured by Western blotting. (**C**) Skin tissue levels of inflammatory cytokines, including IL-17, IL-22, TNF-α, IFN-γ, IL-6, and VEGF, were measured. Data are expressed as means ± SDs (*n* = 6 per experiment). * *p* < 0.05, ** *p* < 0.01, *** *p* < 0.001 vs. the IMQ group. NC: normal control, IMQ: imiquimod, RPE: *Rheum palmatum* L. extract, CP: clobetasol propionate, PCNA: proliferating cell nuclear antigen, and VEGF: vascular endothelial growth factor.

**Figure 3 ijms-23-16000-f003:**
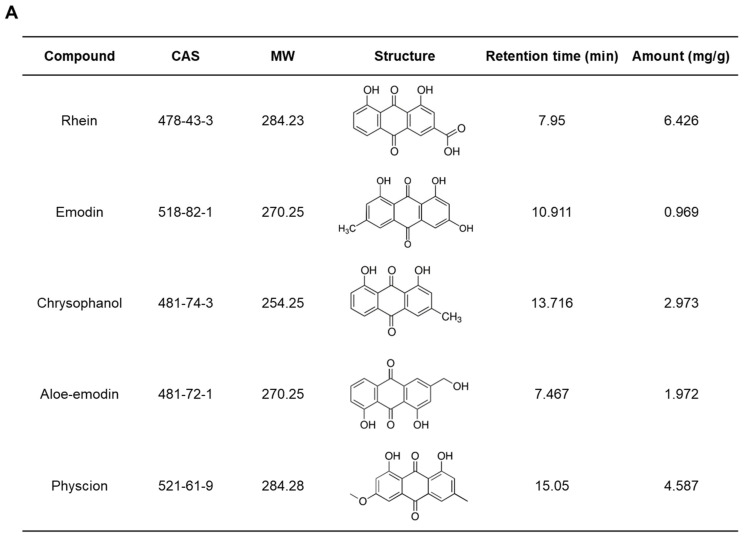
Primary constituents of RPE as determined by HPLC. (**A**) Chemical composition of RPE. (**B**) Chromatograms of the standard compounds (**B**) and RPE (**C**).

**Figure 4 ijms-23-16000-f004:**
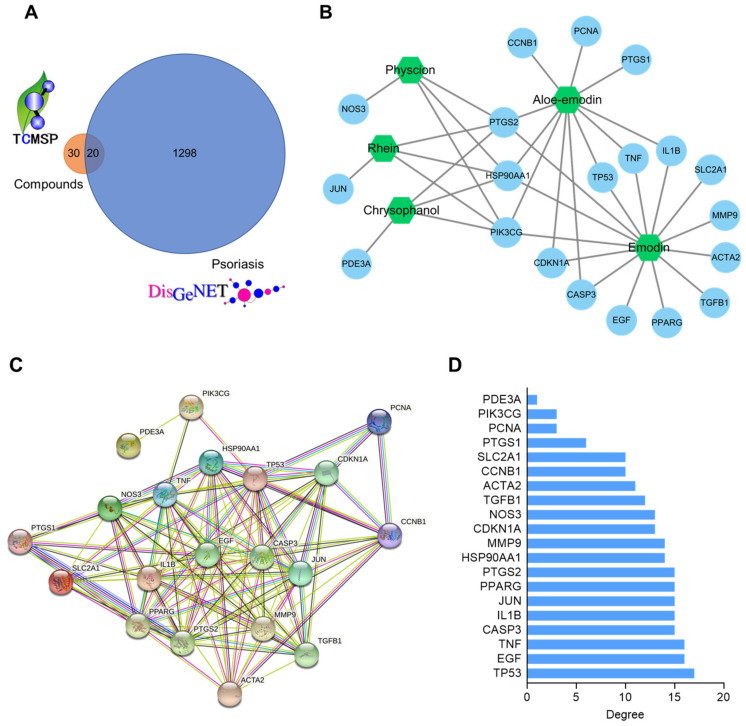
Compound–target and PPI network construction. (**A**) Venn diagram of the targets of RPE components and psoriasis. (**B**) Network of RPE compounds and psoriasis-related targets. Hexagons and circles represent compounds and targets, respectively. The edges are the compound–target interactions. (**C**) The PPI network of psoriasis-related targets. (**D**) Node degree of targets in PPI network.

**Figure 5 ijms-23-16000-f005:**
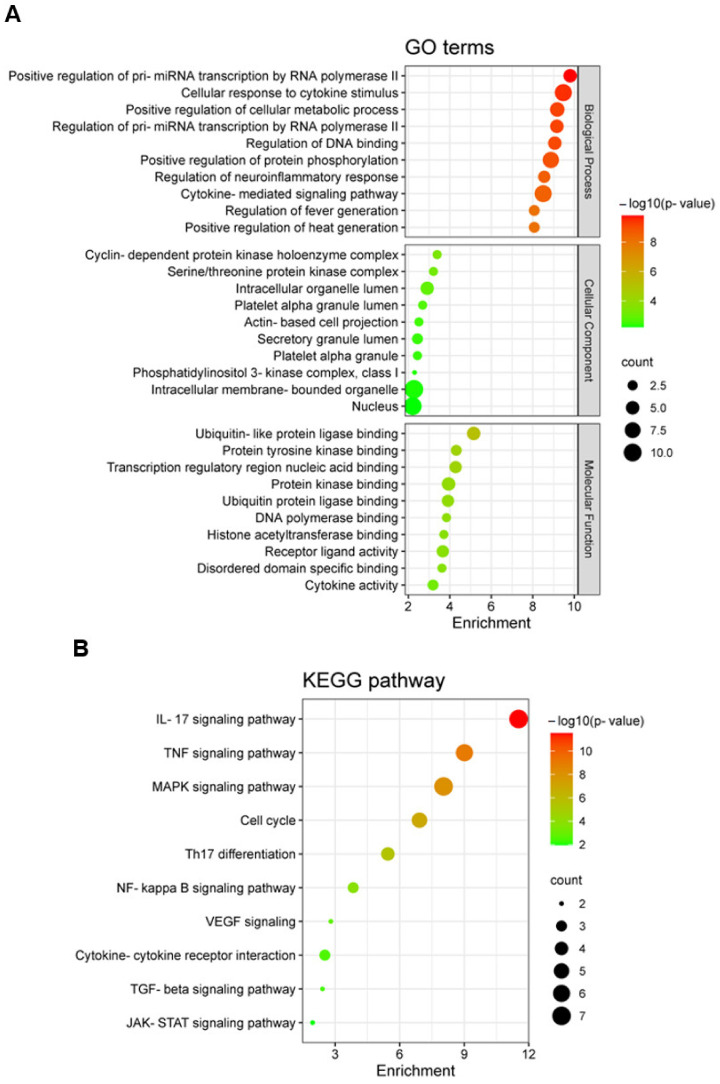
GO and KEGG enrichment analysis. (**A**) Bubble plot of GO enrichment analysis. (**B**) Bubble plot of KEGG pathway enrichment analysis. (**C**) IL-17 signaling pathway was constructed using the KEGG mapper. Blue represents the potential targets of the RPE components.

**Figure 6 ijms-23-16000-f006:**
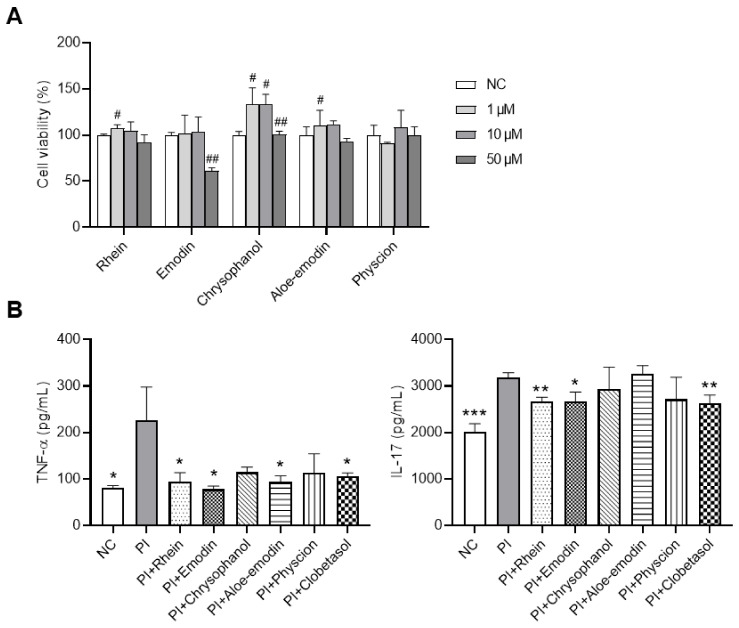
Effects of RPE compounds on the inflammatory response in PI-stimulated EL-4 cells. (**A**) EL-4 cell viability was assessed using XTT assays. (**B**) Effects of RPE compounds on the production of TNF-α and IL-17 in EL-4 cells stimulated with PI (10 ng/mL PMA and 100 ng/mL ionomycin). Data are expressed as means ± SDs (*n* = 3 per experiment). # *p* < 0.05, ## *p* < 0.01 vs. normal control (NC); * *p* < 0.05, ** *p* < 0.01, *** *p* < 0.001 vs. PI-treated cells.

**Figure 7 ijms-23-16000-f007:**
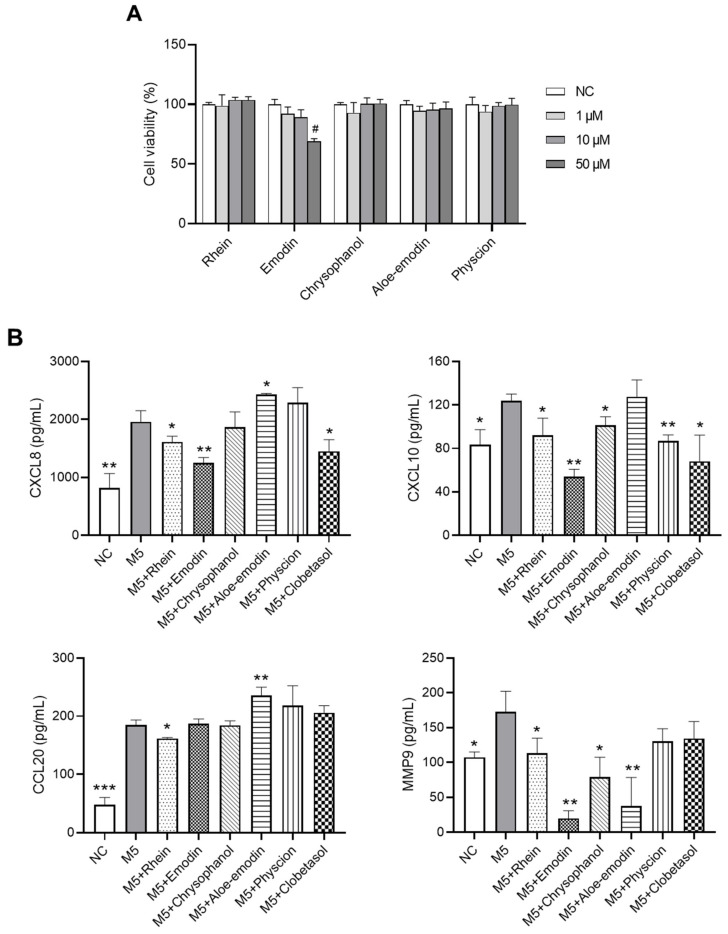
Effects of RPE compounds on the inflammatory response and proliferation in M5-stimulated HaCaT cells. (**A**) HaCaT cell viability was evaluated using XTT assays. (**B**) Effect of RPE compounds on the production of CXCL8, CXCL10, CCL20, and MMP9 in HaCaT cells stimulated with M5 (TNF-α, oncostatin M, IL-1α, IL-17A, and IL-22, 10 ng/mL each) was examined by ELISA assays. (**C**) The effect of RPE compounds on hyperproliferation in M5-stimulated HaCaT cells was measured using the Muse Ki67 Proliferation kit. Data are expressed as means ± SDs (*n* = 3 per experiment). # *p* < 0.05 vs. normal control (NC); * *p* < 0.05, ** *p* < 0.01, *** *p* < 0.001 vs. M5-treated cells. CXCL: chemokine (C-X-C motif) ligand, CCL: CC chemokine ligand, and MMP: matrix metalloproteinase.

**Figure 8 ijms-23-16000-f008:**
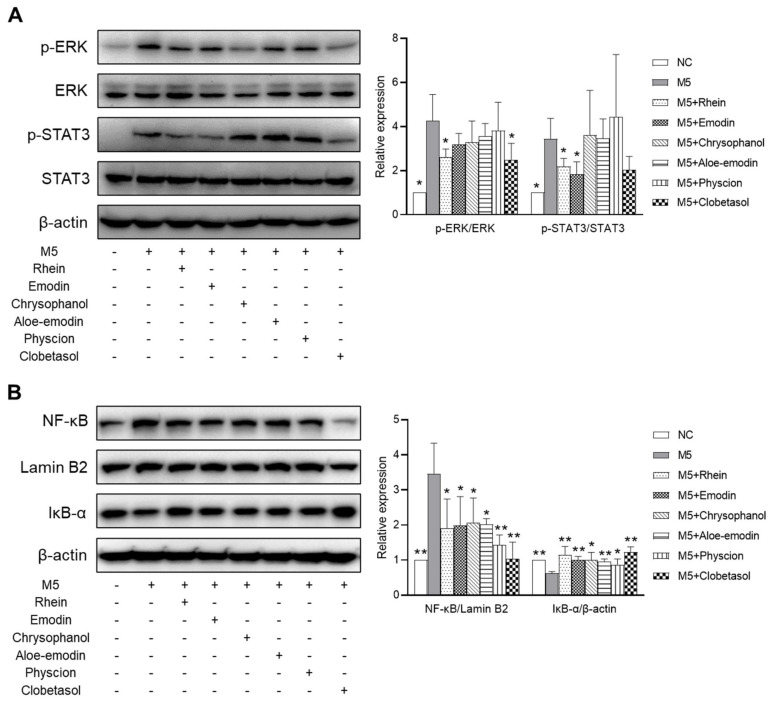
Effects of RPE compounds on ERK, STAT3, and NF-κB pathways in M5-stimulated HaCaT cells. The cells were pretreated with RPE compounds (10 μM) for 1 h and then stimulated with M5 for 30 min. (**A**) Proteins were analyzed by Western blot and the band intensities of p-ERK and p-STAT3 were normalized to ERK and STAT3, respectively. (**B**) Proteins were analyzed by Western blot and the band intensities of NF-κB (nuclear fraction) and IκB-α (cytoplasmic fraction) were normalized to lamin B2 and β-actin, respectively. Data are expressed as means ± SDs (*n* = 3 per experiment). * *p* < 0.05, ** *p* < 0.01 vs. M5-treated cells. ERK: extracellular signal-regulated kinase, STAT3: signal transducer and activator of transcription 3, NF-κB: nuclear factor kappa B, and IκB-α: NF-κB inhibitor alpha.

## Data Availability

The data presented in this study are available in this article.

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
