# Peer review of "Anti-Psoriatic Effect of Rheum palmatum L. and Its Underlying Molecular Mechanisms"

_ijms, 2022, doi:10.3390/ijms232416000_

Round 1

Reviewer 1 Report

It is clear that the authors worked very hard, both in the lab and on the pc, and i would like to aknowledge this.

The experiments are clearly conducted and presented but the text has some problems.

There are too many acronyms that are not defined or are defined at the end, which is annoying for the reader. (eg. coscam score, CP, PCNA, ERK, STAT, VEGF, CAS, CXCL etc.) Not everyone is familiar with all those acronyms. Also, what are BALB/c mice?

Figure 1, table S2 and Figure 7 on page 18, are not fully visible, but i suppose that is a matter of editing.

Line 48: "owing to their effectiveness and safety" .Who says so? In line 252 effectiveness and safety are mentioned again, with 2 references. Ref 12 is 10 years old, from a journal that i couldn't find many info about. Ref 17 says in the abstract, among others, " For the multi-herb formulations, benefits of oral herbal medicines were shown in several studies, however, a number of these studies are not controlled trials, a diversity of interventions are tested and there are methodological issues in the controlled studies." In Line 53 "remarkable decrease in disease severity when compared to traditional anti-psoriatic pharmacotherapy" So, according to authors and ref. 8 and 9, who are 20 years old, is R. palmatum better than cyclosporine, methotrexate etc? Im not saying that it isn't, but i also don't see clear evidence that it is. So, in my opinion, these lines are exacerbating.

Line 60: "blood stasis commonly occurs in the process of psoriasis". It doesn't, according to western medicine. The herb may be beneficial, for reasons analysed later in this paper, but this benefit has nothing to do with "blood stasis", a pseudo-scientific and "mystical" consept. In the same phrase, ref 13 is not evidence for anything, it's a protocol for a RCT, that has no results posted in clinicaltrials.gov

Line 98: The effect of clobetasol should be mentioned, for fairness. Also in Line 263 a positive for R. palmatum comparison with clobetasol is mentioned, but nowhere in the text better for clobetasol results are mentioned.

Line 264 is potentially dangerous for patients who would read the paper, and it should be changed. The experiments are done in mice, not in humans. Isn't the plant dangerous for pregnant women? Aren't the leaves toxic?

Author Response

[Comment 1]: There are too many acronyms that are not defined or are defined at the end, which is annoying for the reader. (eg. coscam score, CP, PCNA, ERK, STAT, VEGF, CAS, CXCL etc.) Not everyone is familiar with all those acronyms. Also, what are BALB/c mice?

Response: Thank you for pointing this out. We added the full name for these abbreviations.

[Comment 2]: Figure 1, table S2 and Figure 7 on page 18, are not fully visible, but i suppose that is a matter of editing.

Response: Thank you for your comment. We adjusted the figures and tables to display them fully in the manuscript.

[Comment 3]: Line 48: "owing to their effectiveness and safety" .Who says so? In line 252 effectiveness and safety are mentioned again, with 2 references. Ref 12 is 10 years old, from a journal that i couldn't find many info about. Ref 17 says in the abstract, among others, " For the multi-herb formulations, benefits of oral herbal medicines were shown in several studies, however, a number of these studies are not controlled trials, a diversity of interventions are tested and there are methodological issues in the controlled studies." In Line 53 "remarkable decrease in disease severity when compared to traditional anti-psoriatic pharmacotherapy" So, according to authors and ref. 8 and 9, who are 20 years old, is R. palmatum better than cyclosporine, methotrexate etc? Im not saying that it isn't, but i also don't see clear evidence that it is. So, in my opinion, these lines are exacerbating.

Response: Thank you for your suggestion. We replaced the old references with more updated and related references, as well as rewrote the sentences to emphasize the background of the study.

[Comment 4]: Line 60: "blood stasis commonly occurs in the process of psoriasis". It doesn't, according to western medicine. The herb may be beneficial, for reasons analysed later in this paper, but this benefit has nothing to do with "blood stasis", a pseudo-scientific and "mystical" consept. In the same phrase, ref 13 is not evidence for anything, it's a protocol for a RCT, that has no results posted in clinicaltrials.gov

Response: Thank you for your comment. We removed ref. 13 and added more updated and related references, as well as rewrote the sentences to emphasize the background of the study.

[Comment 5]: Line 98: The effect of clobetasol should be mentioned, for fairness. Also in Line 263 a positive for R. palmatum comparison with clobetasol is mentioned, but nowhere in the text better for clobetasol results are mentioned.

Response: Thank you for your comment. We added clobetasol results in the text as suggested.

[Comment 6]: Line 264 is potentially dangerous for patients who would read the paper, and it should be changed. The experiments are done in mice, not in humans. Isn't the plant dangerous for pregnant women? Aren't the leaves toxic?

Response: Thank you for your comment. We rewrote the sentence as suggested.

Reviewer 2 Report

Dear Authors,

The manuscript entitled "Anti-psoriatic effect of Rheum palmatum L. and its underlying molecular mechanisms" reports the investigation of the anti-psoriatic mechanism of R. palmatum. In vitro, in silico, and in vitro experiments have been performed to elucidate the molecular mechanism. 

The study design has been well-planned and executed, and the manuscript is well-organized and well-written. The introduction is concise and provides sufficient context about the topic to the reader. The methodology is clearly described and the results are, in general, well presented. However, some minor adjustments are needed before the manuscript is published:

1. Please review the number of the ethics protocol: lines 351 and 494 do not match. 

2. Figures 1E, 1F, and 6B do not display correctly, please adjust the figures in the document.

3. line 120: Please review the citation of Chromatograms of standards and RPE in line 120. Should be Figures 3B and 3C instead of 3A and 3B. 

4. Table S2 does not fit in the document, please adjust it. 

5. Figure 5C is hardly readable, a slightly bigger font will aid to read the IL-17 signaling pathway. 

6. Figure 6: line 195: please include what M5 stands for in the legend of Figure 6. 

Kind regards,

Author Response

[Comment 1]: Please review the number of the ethics protocol: lines 351 and 494 do not match.

Response: Thank you for your comment. We corrected this mistake.

[Comment 2]: Figures 1E, 1F, and 6B do not display correctly, please adjust the figures in the document.

Response: Thank you for your comment. We adjusted these figures to display them fully in the manuscript.

[Comment 3]: line 120: Please review the citation of Chromatograms of standards and RPE in line 120. Should be Figures 3B and 3C instead of 3A and 3B.

Response: Thank you for your comment. We corrected this mistake.

[Comment 4]: Table S2 does not fit in the document, please adjust it.

Response: Thank you for your comment. We adjusted the table to fit the document.

[Comment 5]: Figure 5C is hardly readable, a slightly bigger font will aid to read the IL-17 signaling pathway.

Response: Thank you for your comment. We modified the figure to make it more readable.

[Comment 6]: Figure 6: line 195: please include what M5 stands for in the legend of Figure 6.

Response: Thank you for your comment. This is our mistake when drawing the graphs. Here, M5 should be PI and we corrected it in Figure 6.